# Effects of Lipids from Soybean Oil or Ground Soybeans on Energy Efficiency and Methane Production in Steers

**DOI:** 10.3390/ani15030321

**Published:** 2025-01-23

**Authors:** Elizabeth Fonsêca Processi, Tiago Cunha Rocha, Laila Cecília Ramos Bendia, Clóvis Carlos Silveira Filho, Alexandre Berndt, Elon Souza Aniceto, Tadeu Silva de Oliveira

**Affiliations:** 1Experimental Campus, Universidade Federal Rural do Rio de Janeiro, Av. Lourival Martins Beda, s/n, Campos dos Goytacazes 28022-560, RJ, Brazil; elizabethufrrj@gmail.com; 2Center for Agricultural Sciences, Universidade Estadual da Região Tocantina do Maranhão, R. Godofredo Viana, 1300, Imperatriz 65900-000, MA, Brazil; tiagoticuro@yahoo.com.br; 3Laboratory of Animal Science, Universidade Estadual do Norte Fluminense Darcy Ribeiro, Av. Alberto Lamego, Campos dos Goytacazes 28013-602, RJ, Brazil; lailabendia@gmail.com (L.C.R.B.); cloviscsfilho@hotmail.com (C.C.S.F.); elon1995@hotmail.com (E.S.A.); 4EMBRAPA Sudeste, Rod. Washington Luiz, Km 234, Fazenda Canchim, São Carlos 13560-970, SP, Brazil; alexandre.berndt@embrapa.br

**Keywords:** beef cattle, energy partitioning, mitigation, oil

## Abstract

Incorporating fats into the diets of ruminants, like beef cattle, can play a significant role in reducing methane emissions, which are a major contributor to climate change. Ruminants naturally produce methane during digestion, mainly through a process called enteric fermentation. In this study, we altered the fermentation process by adding fats to the diet, resulting in reduced methane production. Adding dietary fats can alter this fermentation process, lowering methane production. Moreover, a fat-enhanced diet can boost animal health, reduce disease risks, and promote overall well-being. Lowering methane emissions is crucial for climate action since methane is significantly more potent than carbon dioxide in the short term. Farmers can achieve a practical solution that benefits both livestock productivity and environmental sustainability by focusing on the inclusion of fats in ruminant diets. This strategy helps to meet agricultural goals and contributes positively to combating climate change, making it a beneficial approach for farmers and the planet.

## 1. Introduction

Energy is the most limiting nutrient in meat production for weight gain and finishing carcasses. This limitation is primarily observed in pasture-based production systems, where intake is insufficient due to the low energy value of most forages [1]. Although energy is vital, it is considered a secondary nutrient, with a greater emphasis on correcting protein deficiencies in tropical forage. Using supplements can eliminate deficiencies caused by forages, increase the animals’ weight gain, and maximize profits from the activity [1].

In this context, the success of beef cattle production (finishing carcasses) depends on integrating various technologies and management practices to improve the animals’ ability to produce meat more profitably. As the consumer market evolves and demands higher meat quality, Brazilian beef farms are increasingly focusing on enhancing management practices, reducing costs, and boosting production efficiency [2].

Given this, energy is essential to sustain all vital body processes, and its deficiency is shown through a lack of growth, reproductive failure, and the loss of body reserves, which reduces animal productivity [3]. Energy intake has the most significant influence on the growth rate of meat animals. Diets that minimize energy losses during digestion and increase energy retention lead to better feed utilization efficiency in cattle [4].

Thus, including lipids to replace part of the starch in the diets of high-performance animals to meet their high energy needs is essential to avoid nutritional disorders caused by changes in the fermentation pattern (lipids increase the concentrations of glycerol [via lipolysis] and propionate, both of which are gluconeogenic precursors) [5]. Replacing starch with lipid sources (soybean oil or ground soybean seeds) in ruminant diets can be an effective strategy to improve energy efficiency due to the reduction in methane production and the increase in energy concentration, as lipids provide 2.25 times more calories per gram compared to starch [6]. Additionally, biohydrogenation acts as a hydrogen (H₂) sink, reducing the supply of H₂ to *Methanogenic archaea*, thus decreasing methane production [7]. However, lipid availability depends on how lipids are processed in the rumen. Soybean oil is composed mainly of triglycerides and is a highly available and rapidly digestible source of lipids, as triglycerides are quickly released in the rumen. Whole soybeans contain lipids within plant cells, partially protected by cell walls. These walls need to be broken down for the lipids to become available, affecting both the biohydrogenation and digestion of the lipids [5]. However, excess lipids in the diets of ruminants have negative impacts on ruminal fermentation. They can reduce microbial activity due to their toxic effects on ruminal microorganisms, particularly cellulolytic bacteria, which are responsible for fiber degradation. This can alter the production of volatile fatty acids (VFAs) and ruminal pH and decrease VFA production, as well as affect the digestion of carbohydrates and fiber. To ensure the health and productivity of ruminants, it is essential to balance the amount of lipids in the diet to optimize ruminal fermentation and energy production [8].

We hypothesized that adding lipids from different processing forms (soybean oil or ground soybean seeds) would affect the energy utilization efficiency of crossbred steers on feedlot diets. Therefore, the objective of this study was to evaluate the energy losses of feedlot steers fed diets based on corn silage with or without the addition of lipids from different processing forms.

## 2. Materials and Methods

The experiment was conducted in Campos dos Goytacazes, RJ, Brazil (21°45′45″ S, 41°17′06″ W, and 13 m a.s.l.). The local climate is classified as Aw, indicating a humid tropical climate with rainy summers and dry winters, according to the Köppen–Geiger classification system [9].

### 2.1. Animals, Experimental Design, and Diets

The Ethics Committee on Animal Use of the Universidade Estadual do Norte Fluminense approved the experiment (Protocol 207/2013).

Eight ruminally cannulated European–Zebu crossbred steers, with an average live weight of 281.25 kg ± 28.74 kg and 401.06 ± 20.48 kg at the beginning and end of the experiment, respectively, were randomly assigned to two balanced 4 × 4 Latin squares. The animals were dewormed and housed in individual pens with feeders and drinkers, where they went through adaptation to the facilities, feeds, and experimental conditions before the beginning of the trial. Each experimental period lasted 21 days, in which the first 14 days were the adaptation of the animals to diets and the last 7 days for samplings, totaling 84 days. Body weight was measured (without fasting) at 8:00 h at the beginning (1 day) and end (21 days) of each period, and the two values were averaged for each period. The steers were assigned to their dietary treatments within each Latin square so that each treatment followed every other treatment only once during the experiment to balance any residual effects.

Four experimental diets were formulated according to the proportions of ingredients shown in Table 1. Corn silage (CS), corn silage and concentrate feed without lipid addition (W/O), corn silage and concentrate feed with 5% lipid addition (soybean oil [SO]), and corn silage and concentrate feed with 5% lipid addition (ground soybean seeds [SS]) were used. The chemical composition of the diets is presented in Table 2.

### 2.2. Feeding and Feed Intake

Feed was offered twice daily, at 8:00 and 16:00 h, in an amount to allow *ad libitum* intake (minimum of 5% orts). Intake was measured daily (8:00 h) by the difference between offered feed and orts. The feed supply was sampled, and orts were weighed before the morning feeding. Samples corresponding to 10% ort weights were taken and composited per animal per period.

Samples of corn silage, concentrates, and orts were partially dried in a forced air oven at 55 °C for 72 h, processed in a knife mill to 1 mm, and evaluated for dry matter (DM) (AOAC 967.03; [10]) and gross energy content (GE) using an adiabatic bomb calorimeter (Parr 1266, Parr Instruments Co., Moline, IL, USA).

### 2.3. Feces and Urine Collection

Feces was collected and packed in plastic bags. During five consecutive days (15th to 20th experimental day), the feces was collected every two hours, and the total amount of excreted feces was weighed and sampled, making up samples per animal per day and per experimental period. Feces samples were partially dried in a forced air oven at 55 °C for 72 h, processed in a knife mill to 1 mm, and evaluated for dry matter (DM) (AOAC 967.03; [9]) and crude energy content (CE) using an adiabatic bomb calorimeter (Parr 1266, Parr Instruments Co., Moline, IL, USA).

Urine samples were obtained using funnel collectors attached to the animals. Rubber hoses coupled to the hoppers led the urine into plastic containers containing 200 mL of sulfuric acid (H_2_SO_4_) at a 20% concentration. The excreted urine was measured, homogenized, and filtered through filter paper, and 50 mL samples were collected for five consecutive days simultaneously with the feces. These samples were stored at −15 °C for subsequent analysis, composing samples per day, animal, and experimental period. The gross energy content was determined using an adiabatic bomb calorimeter in the urine previously dried in an oven at 55 °C.

### 2.4. Ruminal Methane Emission: Collection of Gases

The daily methane emission was estimated using the tracer gas SF_6_ technique [11], following the methodology described by [12]. Before the experiment, the animals underwent adaptation for gas collection (methane and SF_6_). They were equipped with imitation halters and PVC tanks (yokes) similar in weight and shape to authentic ones but made of inferior material and without devices for gas collection and storage. The animals were evaluated with authentic halters and yokes during the gas collection periods.

Ruminal methane emissions were estimated for five consecutive days. Forty-eight hours before the first evaluation, each animal received an intra-ruminal SF_6_ delivery device, previously standardized for the gas release rate [12]. In the morning (8:00 h) of the first day of each collection period, the animals were individually equipped with the halter and the yoke containing the devices for collecting and storing the gases. Each day (8:00 h), the animals were restrained for the following four days, and the yokes were changed since the devices were calibrated to collect the gases during the 24 h. A baseline yoke was set (1 m high) in the common area of animals in the feedlot to quantify potential methane emissions not coming from the experimental animals. This baseline yoke was also changed daily.

After a collection period of five days, the yokes were sent to the Chemical Ecology Laboratory of Embrapa Meio Ambiente (Jaguariúna-SP) for chromatographic analysis and the determination of the gases sulfur hexafluoride (SF_6_) and methane (CH_4_). The concentrations of the gases CH_4_ and SF_6_ were determined on a gas chromatograph equipped with two injectors coupled to two automated valves. One valve was the flame ionization detector (FID) for reading methane, and the other was the electron capture detector (μECD) for reading SF_6_. The capillary columns Plot HP-Al/M (for methane) and HP-MolSiv (for SF_6_) were between the injector and the detector.

It was assumed that the emission pattern of SF6 simulated that of CH_4_ and that the quantification of methane gas in the sample was a function of SF_6_ flow emitted by the capsules in the animal [12]. The calculation of the emission rate of CH_4_ (QCH_4_) was based on the concentrations of CH_4_ and SF_6_ measured in the samples and the known rate of release of SF_6_ (QSF_6_) according to Equation (1):(1)$$Q_{CH4}=Q_{SF6}\times \left[{CH}_{4}\right]/\left[{SF}_{6}\right]$$

Basal concentrations of CH_4_ and SF_6_ were subtracted from their concentrations in the baseline yokes. The basal concentrations of SF_6_ were typically very low and could be neglected. However, the CH_4_ (approximately 2 mg/L (ppm); [CH_4_]_b_) needed to be subtracted from the measured concentrations in the yokes of experimental animals ([CH_4_]_y_), and for that, we used a baseline yoke collecting ambient air, according to Equation (2):(2)$$Q_{CH4}=Q_{SF6}\times \left({\left[{CH}_{4}\right]}_{y}-{\left[{CH}_{4}\right]}_{b}\right)/\left[{SF}_{6}\right]$$

### 2.5. Energy Calculations

The intakes of dry matter (DMI), gross energy (GEI), digestible energy (DEI), and metabolizable energy (MEI) were calculated by the following equations proposed by [13,14]:(3)$$DMI\left(kg\right)=DM\;of\;diet\;(kg)-DM\;of\;orts\;(kg)$$(4)$$GEI\left(Mcal\right)=GE\;of\;diet\;(Mcal)-GE\;of\;orts\;(Mcal)$$(5)$$DEI\left(Mcal/day\right)=GEI-GEF$$(6)$$MEI\left(Mcal/day\right)=DEI-\left(GEU+GEM\right)$$
where GEF is the gross energy of feces, GEU is the gross energy of urine, and GEM is the gross energy of methane. The energy loss of 0.0133 Mcal/g of CH4 was assumed to quantify the energy lost in the form of methane [15].

The concentrations of gross energy [GE], digestible energy [DE], and metabolizable energy [ME] in the diet, expressed as Mcal/kg DM, were obtained according to Equation (7):(7)$$Concentration\;of\;energy\;in\;diet\;\left(Mcal/kg\right)=energy\;intake(Mcal)/DMI(kg)$$

The contents of net energy for maintenance (NEm) and net energy for gain (NEg) in diets were estimated according to the equations adopted by [13]:(8)$$NEm\left(Mcal/day\right)=1.37\;ME-0.138\;{ME}^{2}+0.0105\;{ME}^{3}-1.12$$(9)$$NEg\left(Mcal/day\right)=1.42\;ME-0.174\;{ME}^{2}+0.0122\;{ME}^{3}-1.65$$

### 2.6. Statistical Analysis

In the statistical analysis of data, the following model was used:$$y_{ijkl}=\mu +\alpha_{i}+\beta_{j}+\gamma_{k}+\tau_{l}+e_{ijkl}$$
where *Y_ijkl_* is the observation of Latin Square *l* on animal *k* in period *j* under treatment *i*; α_1_ is the fixed effect of the *i*-th treatment, *i* = 1, 2, 3, and 4; *β_j_* is the random effect of the *j*-th period, *j* = 1, 2, 3 and 4; *γ_k_* is the random effect of the *k*-th animal, *k* = 1, 2, 3, 4, 5, 6, 7, and 8; *τ_l_* is the random effect of the *l*-th Latin square *l*= 1 and 2; and *e_ijkl_* is the random error associated with each observation, assumed to be normal and independently distributed with mean zero and variance σ^2^.

The statistical model was fit using the PROC MIXED procedure of SAS (SAS OnDemand for academics), as estimated by the maximum likelihood method and the matrices of variance–covariance arranged as composite symmetry, auto-regressive correlation, and random auto-regressive correlation [16]. The choice of covariance structure was made using the Akaike criterion (${AIC}_{c}$) [17,18].

After conducting the analysis of variance, the treatment sum of squares was partitioned into three orthogonal contrasts: C1—comparing animals that did not receive a concentrate supplement (CS) with those that received a concentrate supplement (W/O, SO, and SS); C2—comparing animals that received a supplement without lipids (W/O) with those that received a supplement with lipids (SO and SS); C3—comparing animals supplemented with soybean oil (SO) with those supplemented with soybean grains (SS).

## 3. Results

We observed that treatment had a significant effect (*p* < 0.05) on all variables except for gross energy in feces (GEF). Animals that were fed only corn silage (CS) exhibited lower (*p* < 0.001) dry matter intake (DMI) compared to those on diets with concentrate supplements (Table 3). The gross energy intake (GEI) also showed significant differences *(p* = 0.003), as did the gross energy in feces (GEF) (*p* = 0.016), reflecting similar trends to DMI. Additionally, the digestible energy intake (DEI) was higher (*p* = 0.002) in animals fed diets with concentrate supplements (Table 3). However, the gross energy in urine (GEU), as well as GEU expressed as a percentage of GEI and DEI, was also higher (*p* < 0.05) in those same animals. In our analysis of methane, we found that the addition of lipids significantly reduced (*p* < 0.05) gross energy in methane (MGE), as well as MGE expressed as a percentage of both GEI and DEI (Table 3). Furthermore, animals on concentrate supplement diets showed increases (*p* < 0.05) in metabolizable energy intake (MEI) and MEI as a percentage of GEI. MEI as a percentage of DEI increased even more significantly (*p* < 0.001) with the inclusion of lipids (Table 3).

Methane production (CH_4_) was not significantly affected by the lipid processing form in the diet (*p* > 0.05) (Table 4). However, the inclusion of lipids in the diet led to a reduction in methane production (*p* < 0.05) (Table 4). The concentrate supplements had impacts only on CH_4_/MR (*p* = 0.027) and CH_4_/DMI (*p* = 0.016) (Table 4).

Regarding energy partitioning, the form of lipid processing did not show a significant effect (*p* > 0.05) (Table 5). However, animals that were fed lipids in their diet exhibited increased GE (*p* < 0.001), DE (*p* = 0.017), and ME (*p* = 0.013). In contrast, animals fed diets containing concentrate supplements displayed higher energy efficiency (*p* < 0.05) compared to those that received only corn silage (Table 5).

## 4. Discussion

The increased DMI resulting from using concentrate supplements occurs because ruminants can adjust their consumption according to their energy requirements. When they consume a diet rich in concentrates, they can intake more energy without increasing the volume of feed consumed [19]. However, as noted in previous studies [20,21], multiple signals are integrated into the brain’s feeding centers to regulate eating behavior. The liver is a key sensor of the body’s energy status, integrating both short-term and long-term regulatory mechanisms. It is uniquely positioned to detect available energy and balance it with nutrient demands. According to [20,21], increased lipid intake raises the levels of free fatty acids in the blood, which act as peripheral signals influencing the feed intake control centers in the hypothalamus. This phenomenon, however, was not observed in this study. Although not directly measured, it is essential to highlight the role of glucagon in regulating feed intake. With the inclusion of lipids in the diet, glucagon can modulate food intake, as the body interprets this condition as a state of high energy availability, thereby reducing the need for additional feed intake. [21]. The reduction in DM of CS can also be explained by the low CP content of the diet (54 g/kg).

Including concentrate supplements, such as grains rich in starch and protein, which are more energy-dense than roughage feeds like silage, increases dietary energy intake (GEI, DEI, and MEI). However, the decreased fecal energy loss as a proportion of GE intake (GEF [% GEI]) may be due to increased DM digestibility, as concentrates are more digestible than forages, which also corroborates with the findings of [22,23]. Our study observed lower urinary energy loss in animals that received exclusively corn silage compared to those that received concentrate supplements, whether or not lipid was added. According to [24], GEU primarily results from nitrogenous constituents in urine, including urea, purine derivatives, creatine and creatinine, and hippuric acid. The formation of hippuric acid in the liver is driven by the dietary concentration of degradable phenolic acids, typically higher in forages than in concentrates [23,25]. The heat of combustion of hippuric acid is higher than that of urea [26]. Although these changes may be quantitatively small, GEU accounts for approximately one-third to one-half of the energy losses when transitioning from DE to ME [21]. The urinary energy losses, expressed as a percentage of gross energy intake (UGE [% GEI]) and digestible energy intake (UGE [% DEI]), were higher in animals that received concentrate supplements, with or without lipid addition, compared to those fed exclusively corn silage. This difference is likely attributed to the higher nitrogen contents of the supplemented diets. Interestingly, the addition of lipids to the concentrate and the method of lipid processing did not influence urinary energy losses. This may be because the lipid inclusion did not impact DMI or alter the protein contents of the diets. It is important to note that the availability and processing of lipids in the rumen differ significantly. Soybean oil contains more rapidly digestible lipids, providing immediate availability, while whole soybeans release lipids more gradually and require more processing for complete digestion. The decision to use soybean oil or whole soybeans should be based on the specific needs of the feeding system.

Methane is a byproduct of ruminal fermentation, produced autotrophically by *methanogenic archaea* from carbon dioxide (CO2) and hydrogen (H2) generated during the fermentation of carbon sources. This methane production represents a loss of gross energy from the diet. In ruminants, the production of enteric gas is influenced by various dietary factors, including the type of carbohydrate, forage processing methods, fat supplementation, and the addition of ionophores, among others [27,28]. Including dietary fat in the diet can reduce methane production in the rumen by decreasing hydrogen accumulation. This reduction occurs through the biohydrogenation of fatty acids, which lowers the intake of fermentable organic matter, reduces fiber digestion, and inhibits the activity of ruminal methanogens and protozoa [7,11]. The inhibitory effect of fats on methane production depends on factors such as their concentration, type, and fatty acid composition and the overall nutrient composition of the diet [7]. In the current study, we observed that adding lipids reduced methane production, decreasing energy losses associated with methane production. It is worth pointing out that incorporating lipids into ruminant diets can reduce feed intake, potentially impacting animal productivity [5]. However, in this study, supplemental lipids in concentrate feed did not diminish feed intake compared to animals fed the concentrate without added lipids. This finding suggests that lipids directly contribute to the reduction in methane emissions. Specifically, adding lipids to the concentrate led to a 17.8% decrease in energy lost as methane compared to animals receiving the concentrate without any added lipids.

Smaller concentrations of GE, DE, and ME were found in the diet consisting only of corn silage (CS) compared to diets with concentrate supplements (W/O, SO, and SS). The higher GE contents in diets that included concentrate supplements can be attributed to their greater amounts of protein and fat, which have caloric equivalents of 5.6 and 9.4 Mcal/kg, respectively, higher than those of carbohydrates [29]. The increased DE in diets with lipid additions compared to those without can be explained by the superior digestibility of lipids, which exceeds that of carbohydrates. Additionally, the higher ME concentration in diets with added lipids (SO and SS) compared to those without lipid addition (W/O) could be related to reduced energy losses in the form of methane.

The ME provides more valuable information about a diet’s energy content and animals’ energy requirements than DE. This is because ME considers energy losses through urine and methane in addition to fecal losses. Therefore, ME offers a better estimate of the dietary energy available to the animal. A factor of 0.82 was established in studies primarily conducted at maintenance levels of DMI for cattle on high-forage diets. This value has been used by the Nutrient Requirements of Beef Cattle (NRC 1976, 1984, and 2000). According to [30], this ratio can vary significantly based on factors such as intake, animal age, and feed source. Similarly, Ref. [14] indicates that recent data show a variable relationship between ME and DE, ranging from 0.82 to over 0.95, depending on cattle age, intake level, and diet composition [22]. In a recent study, we observed that with the inclusion of lipids, this ratio was 0.92.

## 5. Conclusions

Adding lipids to the diet (5%) reduces energy losses through methane emissions, increasing steers’ energy efficiency. This addition decreases enteric methane production in these animals. Additionally, the way in which the lipids are processed does not impact energy partitioning.

## Figures and Tables

**Table 1 animals-15-00321-t001:** The proportions of ingredients in the experimental diets, expressed as % of DM.

Ingredients	Lipid Sources			
	CS	W/O	SO	SS
Corn silage	100.00	70.00	70.00	70.00
Soybean seeds	-	-	-	16.43
Corn	-	16.88	13.12	13.02
Soybean oil	-	-	3.10	-
Limestone	-	0.59	0.57	0.55
Soybean meal	-	12.53	13.21	-

DM: Dry matter; CS: corn silage diet; W/O: corn silage and concentrate without lipid addition; SO: corn silage and concentrate with 5% lipid addition, supplemented with soybean oil; SS: corn silage and concentrate with 5% lipid addition, supplemented with ground soybean seeds.

**Table 2 animals-15-00321-t002:** Chemical compositions of experimental diets.

Composition	Treatments (g/kg DM)			
	CS	W/O	SO	SS
DM (g/kg of as-fed)	316.5	390.4	389.6	385.0
OM	917.4	930.5	927.2	930.0
CP	54.0	116.5	114.0	113.0
EE	15.2	18.4	47.8	48.4
Ashes	82.6	69.5	72.94	70.0
NDF	521.1	390.4	404.4	383.9

CS: corn silage diet; W/O: corn silage and concentrate without lipid addition; SO: corn silage and concentrate with 5% lipid addition, supplemented with soybean oil; SS: corn silage and concentrate with 5% lipid addition, supplemented with ground soybean seeds; DM: dry matter; OM: organic matter; CP: crude protein; EE: ether extract; and NDF: neutral detergent fiber; all were expressed as g/kg, except DM, which was expressed as-fed.

**Table 3 animals-15-00321-t003:** Energy intake and losses in steers supplemented or not with lipids from different sources (means ± standard error).

Variable	Treatments				*p*-Values		
	CS	W/O	SO	SS	C1	C2	C3
DMI	7.29 ± 0.33	8.54 ± 0.33	8.44 ± 0.33	8.62 ± 0.33	<0.001	0.958	0.331
GEI	28.58 ± 1.34	32.98 ± 1.34	32.99 ± 1.34	33.89 ± 1.34	0.003	0.753	0.600
GEF	9.78 ± 0.51	9.65 ± 0.51	9.19 ± 0.51	9.93 ± 0.51	0.504	0.850	0.169
GEF (%GEI)	34.15 ± 1.78	29.56 ± 1.78	28.12 ± 1.78	29.39 ± 1.78	0.016	0.665	0.559
DEI	18.78 ± 1.13	23.33 ± 1.13	23.80 ± 1.13	23.96 ± 1.13	0.002	0.705	0.924
UGE	0.40 ± 0.06	0.67 ± 0.06	0.65 ± 0.06	0.56 ± 0.06	0.008	0.460	0.323
UGE (%GEI)	1.39 ± 0.18	2.04 ± 0.18	1.97 ± 0.18	1.66 ± 0.18	0.003	0.892	0.074
UGE (%DEI)	2.03 ± 0.25	2.68 ± 0.25	2.84 ± 0.25	2.40 ± 0.25	0.022	0.807	0.156
MGE	1.81 ± 0.18	2.38 ± 0.18	1.87 ± 0.18	1.86 ± 0.18	0.261	0.025	0.985
MGE (%GEI)	6.37 ± 0.67	7.08 ± 0.67	5.71 ± 0.67	5.67 ± 0.67	0.487	0.001	0.922
MGE (%DEI)	9.69 ± 1.0	10.07 ± 1.0	7.95 ± 1.0	8.12 ± 1.0	0.317	0.027	0.935
MEI	16.57 ± 1.07	20.27 ± 1.07	21.27 ± 1.07	21.54 ± 1.07	0.003	0.422	0.871
MEI (%GEI)	58.02 ± 2.07	6136 ± 2.07	64.21 ± 2.07	63.26 ± 2.07	0.002	0.122	0.580
MEI (%DEI)	88.25 ± 0.97	86.86 ± 0.97	89.12 ± 0.97	89.78 ± 0.97	0.587	<0.001	0.395

CS: corn silage diet; W/O: corn silage and concentrate without lipid addition; SO: corn silage and concentrate with 5% lipid addition, supplemented with soybean oil; SS: corn silage and concentrate with 5% lipid addition, supplemented with ground soybean seeds; DMI: dry matter intake in kg/day; GEI: gross energy intake in Mcal/day; GEF: gross energy in feces in Mcal/day; DEI: digestible energy intake in Mcal/day; UGE: urine gross energy in Mcal/day; MGE: methane gross energy in Mcal/day; MEI, metabolizable energy intake in Mcal/day; C1: comparison between animals that did not receive a concentrate supplement (CS) and those that received a concentrate supplement (W/O, SO, and SS); C2: comparison between animals that received a supplement without lipids (W/O) and those that received a supplement with lipids (SO and SS); C3: comparison between animals that received a supplement containing soybean oil (SO) and those that received a supplement containing soybean grain (SS).

**Table 4 animals-15-00321-t004:** Methane production by steers supplemented or not with lipids from different sources (means ± standard error).

Variable	Treatments				*p*-Values		
	CS	W/O	SO	SS	C1	C2	C3
CH_4_/day	139.3 ± 8.65	160.2 ± 8.74	131.8 ± 8.64	143.3 ± 8.54	0.472	0.016	0.289
CH_4_/year	50.9 ± 3.16	59.3 ± 3.19	48.0 ± 3.15	52.3 ± 3.12	0.461	0.014	0.274
CH_4_/MR	1.6 ± 0.25	2.2 ± 0.25	1.8 ± 0.25	1.7 ± 0.24	0.027	0.001	0.925
CH_4_/DMI	20.3 ± 2.52	19.7 ± 2.52	16.2 ± 2.52	16.2 ± 2.52	0.016	0.008	0.893

CS: corn silage diet; W/O: corn silage and concentrate without lipid addition; SO: corn silage and concentrate with 5% lipid addition, supplemented with soybean oil; SS: corn silage and concentrate with 5% lipid addition, supplemented with ground soybean seeds; CH_4_/day: methane production in g/kg; CH_4_/year: methane production in kg/year; CH_4_/MR: methane production by metabolic rate (BW^0.75^); CH_4_/DMI: methane production by dry matter intake in g/kg; C1: comparison between animals that did not receive a concentrate supplement (CS) and those that received a concentrate supplement (W/O, SO, and SS); C2: comparison between animals that received a supplement without lipids (W/O) and those that received a supplement with lipids (SO and SS); C3: comparison between animals that received a supplement containing soybean oil (SO) and those that received a supplement containing soybean grain (SS).

**Table 5 animals-15-00321-t005:** Energy partitioning in steers fed different lipid sources (means ± standard error).

Variables	Treatments				*p*-Values		
	CS	W/O	SO	SS	C1	C2	C3
GE	4.19 ± 0.03	4.19 ± 0.03	4.31 ± 0.03	4.39 ± 0.03	0.006	<0.001	0.080
DE	2.76 ± 0.09	2.96 ± 0.09	3.01 ± 0.09	3.01 ± 0.06	<0.001	0.017	0.992
ME	2.43 ± 0.06	2.58 ± 0.06	2.78 ± 0.06	2.78 ± 0.05	<0.001	0.013	0.954
NEm	1.42 ± 0.04	1.59 ± 0.04	1.65 ± 0.04	1.64 ± 0.04	<0.001	0.253	0.927
NEg	0.84 ± 0.04	0.98 ± 0.04	1.04 ± 0.04	1.03 ± 0.04	<0.001	0.259	0.891

CS: corn silage diet; W/O: corn silage and concentrate without lipid addition; SO: corn silage and concentrate with 5% lipid addition, supplemented with soybean oil; SS: corn silage and concentrate with 5% lipid addition, supplemented with ground soybean seeds; GE: gross energy in Mcal/kg DM; DE: digestible energy in Mcal/kg DM; ME: metabolizable energy in Mcal/kg DM; NEm: estimated net energy for maintenance in Mcal/kg DM; NEg: estimated net energy for production in Mcal/kg DM. C1: comparison between animals that did not receive a concentrate supplement (CS) and those that received a concentrate supplement (W/O, SO, and SS); C2: comparison between animals that received a supplement without lipids (W/O) and those that received a supplement with lipids (SO and SS); C3: comparison between animals that received a supplement containing soybean oil (SO) and those that received a supplement containing soybean grain (SS).

## Data Availability

The data that support this study will be shared upon request to the corresponding author.

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
