# Peer review of "Effects of Lipids from Soybean Oil or Ground Soybeans on Energy Efficiency and Methane Production in Steers"

_animals, 2025, doi:10.3390/ani15030321_

Round 1
Reviewer 1 Report
Comments and Suggestions for Authors
The article submitted for review entitled "Energy efficiency and methane production in steers fed lipids with different forms of processing " is consistent with the scope of the journal . The article is interesting and well written, but I have coment to tille
The title of the paper suggests the use of oil with various degrees of processing, meanwhile in the study the source of oil is soybean oil or soybeans, there is no information about the processing of the oil, therefore the title of the work is misleading
Author Response
#Reviewer 1
Comments: The article submitted for review entitled "Energy efficiency and methane production in steers fed lipids with different forms of processing " is consistent with the scope of the journal . The article is interesting and well written, but I have coment to tille.
The title of the paper suggests the use of oil with various degrees of processing, meanwhile in the study the source of oil is soybean oil or soybeans, there is no information about the processing of the oil, therefore the title of the work is misleading.
We understand the reviewer's question. We have replaced the title with " Effects of lipids from soybean oil or ground soybeans on energy efficiency and methane production in steers" to enhance clarity for the reader.
Reviewer 2 Report
Comments and Suggestions for Authors
This manuscript examines the potential of dietary fats, particularly soybean oil or ground soybeans, to mitigate methane emissions in the diet of beef cattle. While the manuscript offers valuable insights, certain aspects require further attention to enhance the quality of the writing.
The title should be revised to include the different forms of lipid processing, namely soybean oil or ground soybeans, as this information is not provided in the existing title, abstract, or introduction. This is a crucial aspect of the study and should be explicitly stated.
One illustrative example is as follows:
Effects of lipid derived from soybean oil or ground soybeans on energy efficiency and methane production in steers.
Introduction: It is essential to provide clarification regarding the specific forms of lipid processing that are being studied (oil or ground seed) and the ways in which they differ from one another. Furthermore, the rationale for the study should be reinforced by discussing how the inclusion of oil or ground seed could potentially result in a change in energy efficiency and methane production.
Discussion: It is necessary to discuss why the processing of lipids did not have a significant impact on the results and to suggest possible reasons for this.
Author Response
#Reviewer 2
This manuscript examines the potential of dietary fats, particularly soybean oil or ground soybeans, to mitigate methane emissions in the diet of beef cattle. While the manuscript offers valuable insights, certain aspects require further attention to enhance the quality of the writing.
The title should be revised to include the different forms of lipid processing, namely soybean oil or ground soybeans, as this information is not provided in the existing title, abstract, or introduction. This is a crucial aspect of the study and should be explicitly stated.
Suggestions accepted. Thank you. We replaced the title to " Effects of lipids from soybean oil or ground soybeans on energy efficiency and methane production in steers".
We have added the forms in which lipids were administered to the animals’ diets in the abstract and introduction.
One illustrative example is as follows:
Effects of lipid derived from soybean oil or ground soybeans on energy efficiency and methane production in steers.
Suggestions accepted. Thank you.
Introduction: It is essential to provide clarification regarding the specific forms of lipid processing that are being studied (oil or ground seed) and the ways in which they differ from one another.
Suggestions accepted. Thank you. We insert it into the text.
Furthermore, the rationale for the study should be reinforced by discussing how the inclusion of oil or ground seed could potentially result in a change in energy efficiency and methane production.
Suggestions accepted. Thank you. We insert it into the text.
Discussion: It is necessary to discuss why the processing of lipids did not have a significant impact on the results and to suggest possible reasons for this.
Suggestions accepted. Thank you. We insert it into the text.
Reviewer 3 Report
Comments and Suggestions for Authors
Comments and Suggestions for Authors
After reviewing the manuscript entitled “Energy efficiency and methane production in steers fed lipids with different forms of processing”, the following suggestions were made it:
Simple Summary
The Simple Summary must be rewritten entirely because, in its current form, it does not contain the minimum requirements established. The Simple Summary must contain background information on the evaluated subject and briefly describe the main findings obtained in the current study. It must also include a conclusion that is easy to understand for non-specialists.
Abstract
Line 1: Please specify what type (specie) of animals.
Line 6: “corn silage and concentrate feed with 5% lipid; and corn silage and concentrate feed with 5% lipid”. Did you use two exactly the same treatments? Please clarify and correct.
Lines 7-11: In some response variables, the description should be improved by adding the significance values ​​with which the presence or absence of significant effects was detected.
Lines 13-14: The conclusion should be more specific and clearly indicate the type of lipid.
Keywords: energy efficiency. This word used as keyword is the same as those previously used in the title of the manuscript. Keywords should be different from those in the title (but related to the topic) to broaden the reach of academic search engines in case the manuscript is later published.
Introduction
Lines 1-6: Authors should specify under which feeding conditions energy is a limiting nutrient.
Lines 7-11: The authors should specify what type of Beef Cattle Production System, Finishing? Dual Purpose? Also, the information in this paragraph is not well related and connected to the previous paragraph as it seems that they are talking about a new problem (which is not correct). Therefore, it should be modified and corrected.
Lines 16-20: Authors should describe which other feed products can improve ruminants' energy status (e.g., gluconeogenic precursors). They should also adequately justify why lipids were evaluated for energy enhancement instead of using other available products. In addition, lipids have been widely used as part of the ruminant diet. Therefore, authors should clarify what new knowledge their manuscript contributes (to justify its possible publication). In addition, previously reported results from studies using lipid sources similar to those used in the present study should be described to provide a brief overview of the previously found effects. These descriptions should include doses, experimental periods, physiological stages, and types of diet (high or low in forage) used in the cited studies.
Line 21: Please delete “In this study,”.
Lines 23-24: Authors should specify which lipid forms were evaluated.
Material and methods
Section 2.3, lines 1-2: The authors must justify with arguments and relevant scientific references why they consider it valid to take stool samples every 24 hours. The methodology normally used and valid worldwide consists of taking samples daily in the morning and in the afternoon. In addition, the authors did not specify how many days they took samples, which could also reduce the validity of the method used and the results obtained.
Section 2.3, Lines 7-13: Authors must justify the use of these methods with relevant and original scientific references.
Sections 2.4 and 2.5: Authors should review and correct the citations used in these two sections because some of them do not contain the correct format.
Section 2.6: Statistical analyses were performed and described correctly.
Results
Lines 1-3: Please delete these lines.
Lines 8-9: Please delete these lines.
3. Results, Tables 3, 4, and 5: The description of the results in Tables 4 and 5 is correct. However, the description in Table 3 is confusing and needs to be clarified. The description should be improved by adding the significance values ​​with which the presence or absence of significant effects was detected. Also, in Tables 3, 4, and 5 superscripts should be added to help clearly identify which treatment means were different from each other.
Discussion
1-8: Authors should discuss the mechanism through which high energy intakes (e.g., using lipid supplements) can modify dry matter intake. This discussion should include the physiological and hormonal regulatory mechanisms, not just the physical mechanisms of rumen filling.
Lines 11, 16, 22, 43, 44, 48, 51, and 70: The Results Tables should not be cited in the discussion section. Therefore, the citation of Tables 3, 4 and 5 should be removed from these lines.
Overall the entire discussion section is well organized and contains acceptable depth. Therefore, I have no additional suggestions other than those mentioned above.
Conclusions
Lines 325-327: The conclusion drawn by the authors is consistent with the title, objectives, and results of the manuscript. Therefore, I have no additional suggestions.
Comments on the Quality of English Language
The quality of the English language is too low and needs to be improved before the manuscript can be considered for possible publication.
Author Response
#Reviewer 3
Simple Summary
The Simple Summary must be rewritten entirely because, in its current form, it does not contain the minimum requirements established. The Simple Summary must contain background information on the evaluated subject and briefly describe the main findings obtained in the current study. It must also include a conclusion that is easy to understand for non-specialists.
Suggestions accepted. Thank you.
Abstract
Line 1: Please specify what type (specie) of animals.
Suggestions accepted. Thank you.
Line 6: “corn silage and concentrate feed with 5% lipid; and corn silage and concentrate feed with 5% lipid”. Did you use two exactly the same treatments? Please clarify and correct.
Yes, the treatments are not based on the amounts of lipids, but rather on the way these lipids are provided. The availability of these lipids to the animal and how they are processed in the rumen differ. Soybean oil provides lipids that are more quickly digestible and have greater immediate availability, while whole soybeans release lipids more gradually and require more processing for complete release.
Lines 7-11: In some response variables, the description should be improved by adding the significance values ​​with which the presence or absence of significant effects was detected.
Suggestions accepted. Thank you.
Lines 13-14: The conclusion should be more specific and clearly indicate the type of lipid.
Suggestions accepted. Thank you.
Keywords: energy efficiency. This word used as keyword is the same as those previously used in the title of the manuscript. Keywords should be different from those in the title (but related to the topic) to broaden the reach of academic search engines in case the manuscript is later published.
Suggestions accepted. Thank you.
Introduction
Lines 1-6: Authors should specify under which feeding conditions energy is a limiting nutrient.
Suggestions accepted. Thank you.
Lines 7-11: The authors should specify what type of Beef Cattle Production System, Finishing? Dual Purpose? Also, the information in this paragraph is not well related and connected to the previous paragraph as it seems that they are talking about a new problem (which is not correct). Therefore, it should be modified and corrected.
Suggestions accepted. Thank you.
Lines 16-20: Authors should describe which other feed products can improve ruminants' energy status (e.g., gluconeogenic precursors). They should also adequately justify why lipids were evaluated for energy enhancement instead of using other available products. In addition, lipids have been widely used as part of the ruminant diet. Therefore, authors should clarify what new knowledge their manuscript contributes (to justify its possible publication). In addition, previously reported results from studies using lipid sources similar to those used in the present study should be described to provide a brief overview of the previously found effects. These descriptions should include doses, experimental periods, physiological stages, and types of diet (high or low in forage) used in the cited studies.
Suggestions accepted. Thank you.
Line 21: Please delete “In this study,”.
Suggestions accepted. Thank you.
Lines 23-24: Authors should specify which lipid forms were evaluated.
Suggestions accepted. Thank you.
Material and methods
Section 2.3, lines 1-2: The authors must justify with arguments and relevant scientific references why they consider it valid to take stool samples every 24 hours. The methodology normally used and valid worldwide consists of taking samples daily in the morning and in the afternoon. In addition, the authors did not specify how many days they took samples, which could also reduce the validity of the method used and the results obtained.
We apologize for the typographical error. In this study, both feces and urine were collected every two hours for five consecutive days, not just for 24 hours. This has been corrected in the text.
Sections 2.4 and 2.5: Authors should review and correct the citations used in these two sections because some of them do not contain the correct format.
Suggestions accepted. Thank you.
Section 2.6: Statistical analyses were performed and described correctly.
Thank you.
Results
Lines 1-3: Please delete these lines.
Suggestions accepted. Thank you.
Lines 8-9: Please delete these lines.
Suggestions accepted. Thank you.
- Results, Tables 3, 4, and 5: The description of the results in Tables 4 and 5 is correct. However, the description in Table 3 is confusing and needs to be clarified. The description should be improved by adding the significance values ​​with which the presence or absence of significant effects was detected.
We do not understand what the confusion is about Table 3. The description is in the last 3 columns of the Table (P-values), the description of each contrast is in the footer of the table.
Also, in Tables 3, 4, and 5 superscripts should be added to help clearly identify which treatment means were different from each other.
We see the need for this, since the P-values ​​are described in the Tables as well as the description of each contrast in the footer of the Tables.
Discussion
1-8: Authors should discuss the mechanism through which high energy intakes (e.g., using lipid supplements) can modify dry matter intake. This discussion should include the physiological and hormonal regulatory mechanisms, not just the physical mechanisms of rumen filling.
Suggestions accepted. Thank you.
Lines 11, 16, 22, 43, 44, 48, 51, and 70: The Results Tables should not be cited in the discussion section. Therefore, the citation of Tables 3, 4 and 5 should be removed from these lines.
Suggestions accepted. Thank you.
Overall the entire discussion section is well organized and contains acceptable depth. Therefore, I have no additional suggestions other than those mentioned above.
Thank you.
Conclusions
Lines 325-327: The conclusion drawn by the authors is consistent with the title, objectives, and results of the manuscript. Therefore, I have no additional suggestions.
Thank you.
The quality of the English language is too low and needs to be improved before the manuscript can be considered for possible publication.
Suggestions accepted. Thank you. We review English with a native speaker.
Reviewer 4 Report
Comments and Suggestions for Authors
Please see attached PDF. Best regards.

Author Response
#Reviewer 4
Same treatment appears twice where the second one should show soybean inclusion
Yes, the treatments are not based on the amounts of lipids, but rather on the way these lipids are provided. The availability of these lipids to the animal and how they are processed in the rumen differ. Soybean oil provides lipids that are more quickly digestible and have greater immediate availability, while whole soybeans release lipids more gradually and require more processing for complete release.
Citation needed.
Suggestions accepted. Thank you.
repalce with 0800h
Suggestions accepted. Thank you.
Lower DMI for the CS cattle was more that likely depressed because the diet was nitrogen deficient. The CS diet was 5.4% CP, were your other diets were 11% CP near these animal's retirement. Hence, I believe some of the reduced DMI associate with the CS diet was because of a CP deficiency.
Suggestions accepted. Thank you. We insert it into the text.
previously defined.
Suggestions accepted. Thank you.
Needs correct, noun missing.
Suggestions accepted. Thank you.
Again, should this not be NASEM [11]?
Yes, NASEM. Thank you.
previously defined for both.
Suggestions accepted. Thank you.
You need to state that this conclusion is only valid within the parameters of the data because we know the adding an exclusive amount of lipids will limit digestion, decrease DMI, and cause digestive upset.
Suggestions accepted. Thank you. We insert it into the text.
Round 2
Reviewer 2 Report
Comments and Suggestions for Authors
The manuscript has undergone significant improvement.
Author Response
#Reviewer 2
Comments and Suggestions for Authors
The manuscript has undergone significant improvement.
Thank you.